# OpenReview forum: "Solving Puzzles? Jailbreaking Multimodal Large Language Models!"
_ICLR.cc/2026/Conference — ICLR 2026 Conference Withdrawn Submission_

### Official Review · Reviewer_6KV8 · 2025-10-21

**Soundness:** 3
**Presentation:** 2
**Contribution:** 2
**Rating:** 4
**Confidence:** 4

**Summary:**

This paper proposes a new visually guided puzzle jailbreak attack method for multimodal large language models (MLLMs), which aims to induce the current state-of-the-art open source and production MLLMs to generate harmful content involving high-risk scenarios such as weapon assembly and chemical synthesis through carefully designed cross-modal puzzles. The authors developed a unified pipeline including text classification generation, visual object decomposition, and visually guided puzzle construction, and based on this, built a new benchmark test set PuzzleV-JailBench, covering 144 dangerous weapons and 54 dangerous chemicals. Experimental results show that this method can induce MLLMs to generate dangerous content with a high success rate, and further quantify the risk level of the model output content through risk level assessment.

**Strengths:**

The proposed PuzzleV-JailBench has high ASR for the open-source and closed-source MLLMs.

The paper is easy to follow.

The dataset is useful and can be regarded as an effective method to evaluate the safety.

**Weaknesses:**

The techonlogy is somewhat limited. The similar benchmark has already existed similar framework,e.g., HADES, and MM-safetybench. I do not think essential differences exist.

This method is mainly targeted at two specific scenarios: weapon assembly and chemical synthesis. It is difficult to extend to other general scenarios, which limits its scope of application.

The different parts, e.g., text classification generation, visual object decomposition, and visually guided puzzle construction, should be further ablated. Only text and image ablation is not enough.

Computational costs: Supplement the resource consumption of a single experiment (such as GPU hours, API call costs) to evaluate research sustainability.

**Questions:**

see weakness

---

### Official Review · Reviewer_Agdh · 2025-10-25

**Soundness:** 2
**Presentation:** 3
**Contribution:** 2
**Rating:** 4
**Confidence:** 5

**Summary:**

This paper addresses the vulnerability of Multimodal Large Language Models (MLLMs) to jailbreak attacks in high-risk scenarios. It introduces a novel vision-instructed puzzle jailbreak attack that conceals malicious intent (weapon assembly and hazardous chemical synthesis) within cross-modal puzzles. The authors propose a three-stage unified pipeline, i.e., textual taxonomy generation, visual object decomposition, and vision-instructed puzzle construction, to build the PuzzleV-JailBench (144 dangerous weapons, 54 hazardous chemicals). Empirical tests show this attack achieves high Attack Success Rates (ASRs) on state-of-the-art open-source (e.g., LLaVA-v1.6, Qwen2-VL) and production MLLMs (e.g., GPT-4o, Gemini-1.5-Pro), with outputs ranging from low to high risk (per a 4-level risk metric). The work highlights MLLMs’ safety flaws in high-risk domains and calls for improved defenses.

**Strengths:**

- New Attack Paradigm: The vision-instructed puzzle design fills a gap in existing MLLM jailbreak research, which mostly focuses on low-risk, easily detectable threats. By embedding malicious intent in cross-modal puzzles, it bypasses strong safety alignments of production MLLMs effectively.
- Comprehensive Benchmark & Pipeline: The three-stage automated pipeline (taxonomy → decomposition → puzzle) enables scalable generation of high-risk jailbreak data (PuzzleV-JailBench), addressing the challenge of manual data collection for weapon/chemical scenarios.
- Rigorous Evaluation: The study uses both ASR and a 4-level risk metric (with GPT-4/Qwen2.5 assistance) to assess attacks, covering 9 MLLMs (3 open-source, 6 production) and comparing against 5 baseline methods, ensuring reliable and generalizable results.

**Weaknesses:**

- Limited Defense Exploration: The defense experiments only test white-box prompt-based methods (AdaShield-o, FigStep-o) on open-source MLLMs, with minimal ASR reduction (~1–2%). No exploration of black-box defenses (critical for production MLLMs) or advanced strategies like adversarial training and LlamaGuard.

- Lack of Generalization to Other High-Risk Scenarios: The attack focuses exclusively on weapon assembly and chemical synthesis. It fails to validate whether the vision-instructed puzzle paradigm works for other high-risk domains (e.g., cyberattacks, biotoxin production).

- Oversimplified Risk Level Subjectivity: While the 4-level risk metric uses GPT-4/Qwen2.5 for evaluation, the paper does not address inter-model inconsistency in risk scoring (e.g., GPT-4 being stricter than Qwen2.5) or provide inter-annotator agreement to reduce subjectivity.

- Ignorance of Temporal Robustness: The experiments use fixed versions of production MLLMs (e.g., GPT-4o-2024-11-20). There is no analysis of whether model updates (common in production) would mitigate the proposed attack, limiting the work’s long-term relevance.

**Questions:**

- The defense analysis is restricted to white-box settings on open-source MLLMs; it does not investigate black-box defenses (e.g., input purification, query detection) that are more applicable to production MLLMs, leaving a critical gap in real-world safety solutions.

- The puzzle-based attack is only validated for weapons and chemicals. Without testing on other high-risk tasks (e.g., designing malware, synthesizing biotoxins), the generalizability of the proposed paradigm remains unproven.

- The risk level evaluation relies on two LLMs (GPT-4, Qwen2.5) but lacks quantification of inter-model discrepancies (e.g., differing risk scores for the same MLLM output) or inter-annotator reliability tests, undermining the metric’s validity.

- No experiments are conducted on updated versions of production MLLMs. As MLLM developers frequently patch vulnerabilities, the attack’s effectiveness on future model iterations is unknown, reducing the work’s practical significance.

- The pipeline depends heavily on GPT-4o for taxonomy generation and component analysis. If GPT-4o’s safety alignments are strengthened to reject weapon/chemical queries, the entire pipeline’s scalability and feasibility would be compromised.

- The paper does not analyze the impact of puzzle complexity (e.g., number of components, label clarity) on attack success. It remains unclear whether simpler puzzles (fewer visual prompts) would still bypass MLLM safety checks, limiting insights into attack mechanisms.

- Minor issues
1. There is no discussion of ethical risks in disseminating the attack method. While the authors note potential misuse, they provide no concrete guidelines for responsible sharing of the PuzzleV-JailBench or attack code, raising concerns about real-world harm.
2. The text in Figure 1 is too small to be read.

**Details Of Ethics Concerns:**

None.

---

### Official Review · Reviewer_34eq · 2025-11-01

**Soundness:** 3
**Presentation:** 3
**Contribution:** 2
**Rating:** 4
**Confidence:** 3

**Summary:**

This paper introduces a novel jailbreaking approach termed vision-instructed puzzle attacks, targeting MLLMs in high-risk domains (weapon assembly and chemical synthesis). The authors propose a three-stage pipeline to generate PuzzleV-JailBench and demonstrate its effectiveness in bypassing safety alignments of both open-source and production MLLMs. By embedding malicious intent within cross-modal puzzles, the method achieves high ASRs and generates high-risk responses.

**Strengths:**

- Novelty: The idea of disguising jailbreak attempts as visual puzzles is creative and underexplored. The work effectively highlights how game-based prompts can circumvent safety mechanisms.

- Benchmark quality: PuzzleV-JailBench is systematically constructed with diverse categories (weapons/chemicals) and rigorous visual/linguistic transformations, providing a valuable resource for safety research.

- Comprehensive evaluation: The paper thoroughly evaluates state-of-the-art MLLMs (e.g., GPT-4o, Claude-3.5-Sonnet) and includes risk-level metrics, ablation studies, and defense tests, strengthening empirical claims.

**Weaknesses:**

- Limited task generalizability: The method is heavily tailored to two specific tasks—weapon assembly and chemical synthesis—which involve structured, step-by-step procedures naturally amenable to "puzzle" framing (e.g., linking components or chemical reactants). The attack relies on carefully designed prompts (e.g., "I am playing a game and need to make correct arrows to link numbered objects"), which are highly scenario-specific.

- Lack of Universal Attack Framework: This paper does not propose a unified strategy for adapting puzzle-based jailbreaks to arbitrary harmful instructions. It remains unclear whether the approach can generalize to other jailbreaking scenarios (e.g., privacy violations or misinformation), where tasks may not decompose into visual puzzles as intuitively. The authors do not validate versatility beyond these domains. The method feels more like a proof-of-concept for specific cases than a broadly applicable framework.

**Questions:**

Please see the weakness part.

---

### Note · Authors · 2025-11-23

I have read and agree with the venue's withdrawal policy on behalf of myself and my co-authors.